# Structural basis for membrane microdomain formation by a human Stomatin complex

Jack Stoner[1,2], Shufang Li[3] & Ziao Fu [1,2] ✉

Biological membranes are not just passive barriers—they actively sense and respond to mechanical forces, in part through specialized proteins embedded within them. Among these are Stomatin-family proteins, which are known to influence membrane stiffness and regulate ion channels, yet how they achieve these functions at the molecular level has remained elusive. Here, we report the 2.2 Å cryo-electron microscopy structure of the human Stomatin complex in a native membrane environment. We find that Stomatin assembles into a 16-subunit ring-shaped homo-oligomer, forming a ~12 nm-wide cage that defines a mechanically distinct, curvature-resistant membrane microdomain. While the majority of the complex exhibits C16 symmetry, the C-terminal domains adopt two alternating conformations, producing a symmetry-broken hydrophobic β-barrel pore with local C8 symmetry. The membrane beneath the complex remains flat despite surrounding curvature, indicating localized membrane stiffening. The structure reveals a conserved network of inter-subunit salt bridges that stabilize the assembly. These findings provide a molecular framework for how Stomatin oligomers shape membrane architecture and mechanics, offering insight into their roles in mechanotransduction and diseases such as nephrotic syndrome.

Stomatin was originally discovered in patients with hereditary stomatocytosis, a rare hemolytic anemia characterized by altered red blood cell shape and increased membrane permeability[1–4]. Since then, Stomatin has been recognized as a widely expressed and evolutionarily conserved membrane protein[5,6], with closely related family members such as Podocin[7], Stomatin-like protein 3 (STOML3)[8], and MEC-2[9], which share high sequence similarity and conserved structural and functional features. Podocin mutations are a major cause of nephrotic syndrome, a pediatric kidney disorder characterized by proteinuria and dysfunction of the glomerular filtration barrier[7,10–13]. In C. elegans, MEC-2 is essential for touch sensitivity[14–16], while STOML3 in mammals regulates the activity of mechanosensitive Piezo channels in somatosensory neurons[17–19]. Collectively, these findings point to a conserved role for stomatin-family proteins in modulating membrane properties and mechanosensitive signaling.

One longstanding hypothesis is that Stomatin and its homologs function as membrane scaffolds, forming oligomeric assemblies that define specialized membrane microdomains—often associated with lipid rafts[20–23]. Biochemical studies have suggested Stomatin's localization to detergent-resistant membrane fractions[24,25] and their involvement in membrane stiffening, cholesterol binding[26], and regulation of membrane signaling complexes[27,28]. Despite these insights, a detailed structural understanding of how Stomatin oligomerizes and modulates membrane mechanics has remained elusive. While previous studies have reported crystal[29,30] and NMR[31] structures of isolated SPFH domains and inferred oligomerization through biochemical assays, high-resolution structures of full-length, membrane-embedded Stomatin complexes have not been available.

In this study, we apply the native membrane vesicle approach to determine the high-resolution cryo-EM structure of full-length human Stomatin in its membrane environment. Our results reveal that

[1]Center for the Investigation of Membrane Excitability Diseases, Washington University School of Medicine, St. Louis, MO, USA. [2]Department of Cell Biology and Physiology, Washington University School of Medicine, St. Louis, MO, USA. [3]Department of Pediatrics, St Louis Children's Hospital, Washington University School of Medicine, St. Louis, MO, USA. ✉e-mail: ziao@wustl.edu

Stomatin assembles into a closed 16-subunit ring that defines a flat, curvature-resistant membrane microdomain. The structure shows how alternating protein conformers form a rigid scaffold through conserved inter-subunit salt bridges and a C-terminal β-barrel, generating a hydrophobic pore at the narrow end of the complex. These findings provide a molecular framework for understanding how Stomatin and its homologs organize membrane architecture, regulate membrane signaling complexes, and contribute to diseases such as nephrotic syndrome.

## Results

### Cryo-EM structure of the human Stomatin complex reveals a 16-mer oligomer

We used a native membrane vesicle method[32] combined with cryo-EM single-particle analysis to study human Stomatin. The protein was overexpressed in human cells with an ALFA tag on the N-terminus and isolated in vesicles (Fig. 1a). Two-dimensional class averages showed that Stomatin forms closed, cap-like structures (Fig. 1b and Supplementary Fig. 1a). Side views of the particles showed that the complex is partially inserted into the membrane. The 3D reconstruction showed a ring-shaped assembly made of 16 Stomatin subunits, anchored to the membrane at the wide end and forming a pore at the narrow end (Fig. 1c). The structure is about 16 nm wide and 10 nm tall, and was resolved to 2.2 Å (Fig. 1d and Supplementary Fig. 1b–d). The ring is made of two alternating forms of the protein, called Conformer A and Conformer B, arranged in a repeating A–B–A–B pattern, resulting in local C8 symmetry within the β-barrel region despite the overall C16 ring architecture. Both conformers share the same domain layout (Fig. 1e and Supplementary Fig. 1e). The N-terminal region is mostly flexible, and only a short region is visible in the map interacting with other domains. Next are two short hydrophobic helices (H1 and H2), followed by the conserved SPFH1 and SPFH2 domains, which form the main structure. The SPFH domains are oriented roughly perpendicular to the membrane. A short helix called the wall helix comes after SPFH2. This helix is shorter than in some other SPFH family members like Prohibitin and Flotillin, but the same in size as Mec-2, Podocin and STOML3. After the wall helix, a loop leads to another helix we call the cap helix, which bends inward near the narrow end. This helix continues into a β-strand, forming a β-barrel made by all 16 subunits. The β-barrel creates a central pore at the narrow end of the complex, about 1.8 nm wide. The key difference between Conformer A and Conformer B is in the C-terminal region. Conformer A ends in a short helix, while Conformer B ends in a loop (Fig. 1f and Supplementary Fig. 1f). This difference adds asymmetry to an otherwise symmetric ring. A short region at the very end of the protein (~10 amino acids) is not visible and may stick out from the complex. A cartoon model shows the arrangement of all domains and how the two conformers are organized in the ring. These results show that human Stomatin forms a closed, 16-subunit ring with built-in asymmetry.

### Stomatin interacts with the membrane via N-terminal helices and the hydrophobic surface of SPFH1 domain

Cryo-EM analysis shows that the Stomatin complex anchors to the membrane through two short hydrophobic helices at its N-terminus—H1 and H2—and a membrane-facing hydrophobic surface on the SPFH1 domain (Fig. 2a). These helices are not transmembrane; instead, they insert halfway into the lipid bilayer without crossing it. The H1 and H2 helices are stabilized by aromatic and hydrophobic residues. Proline 47, located at the junction between H1 and H2, is a key structural element known to influence whether such hairpins remain embedded or convert into full transmembrane helices[33]. Three cysteine residues—C30, C53, and C87—are found in this region and are known sites of S-palmitoylation, which further supports membrane attachment[34]. Among them, C53 corresponds to C124 in human podocin; the disease-linked mutation C124W has been associated with nephrotic syndrome

(Supplementary Fig. 2). Tryptophan 51, also located in H2, contributes to membrane anchoring. In podocin, mutations at the equivalent residue (W122S/L/*)[35] are also associated with nephrotic syndrome. At the interface between the membrane and cytosol, Aspartate 89 and Arginine 67 within the SPFH1 domain form a salt bridge (Fig. 2b). These residues are conserved and important for domain stabilization. Their equivalents in podocin—D160 and R138—are common mutation hotspots (D160G, R138P/Q[36]/*). Additionally, Lysine 55 (K55), located on the hydrophilic face of SPFH1 just below the membrane, corresponds to another disease-associated site in podocin (K126N).

### The Stomatin complex creates a mechanically distinct membrane domain

Several members of the stomatin protein family—including Podocin, Stomatin-like protein 3, and C. elegans MEC-2—have been implicated in modulating membrane stiffness and mechanosensation, with cholesterol binding proposed as a contributing factor. However, our structural analysis of human Stomatin reveals that the only regions in direct contact with the lipid bilayer are the N-terminal helices (H1 and H2) and the hydrophobic face of the SPFH1 domain. We observed no structural features consistent with cholesterol-binding pockets or associated densities in these regions. This prompted us to investigate whether Stomatin may regulate membrane mechanics through an alternative mechanism. Using cryo-EM, we found that vesicles containing Stomatin exhibited a wide range of radii—from ~53 nm to ~12.5 nm (Fig. 2c). Importantly, the Stomatin complex itself maintained a consistent 16-mer oligomeric architecture across all vesicles. Despite this variation, the membrane curvature directly beneath the Stomatin complex remained constant, while the surrounding membrane conformed to the vesicle's overall curvature. To explore this further, we grouped particles based on vesicle size and reconstructed Stomatin complexes from five curvature-defined datasets (Fig. 2d). Across these conditions, the membrane footprint of the complex changed minimally—shrinking by only ~1.9 Å on each side between the largest and smallest vesicles (Fig. 2e). These results suggest that Stomatin oligomers form a mechanically distinct membrane microdomain, capable of resisting external curvature changes. This stabilization is likely conferred by the cage-like architecture of the complex and its influence on local lipid organization, providing a model for how Stomatin proteins contribute to membrane mechanics. These findings reveal that Stomatin oligomers generate a membrane microdomain with distinct mechanical properties—able to maintain its curvature even when embedded in vesicles of widely varying sizes. This remarkable rigidity prompted us to examine what structural features give rise to such stability at the molecular level.

### Conserved salt bridges stabilize the Stomatin oligomer

The Stomatin complex is held together by a network of conserved salt bridges between subunits that span across several structural layers of the oligomer (Fig. 3a). These electrostatic interactions are important for maintaining the stability of the 16-subunit ring. In the SPFH1 domain (layer I), a salt bridge is formed between E59 and R70 from neighboring subunits (Fig. 3b). These residues correspond to known disease-associated positions in podocin—E130K and H141Y—highlighting the importance of this contact for proper assembly. In the wall and cap helices (layers II–IV), additional salt bridges support the narrow-end region. For example, R205 from the wall helix of one subunit forms a salt bridge with E262 in the cap helix region of a subunit two positions away (Fig. 3c). This cross-subunit interaction goes beyond immediate neighbors and adds another layer of stabilization. In the mid-layer of the narrow-end region(layer III), a salt bridge forms between E227 and K220 from adjacent subunits wall helices, while R251 from cap helix from another subunit further stabilizes the interface (Fig. 3d). At the very bottom of the narrow end (layer IV), E243 in the wall helix engages K235 from a neighboring subunit

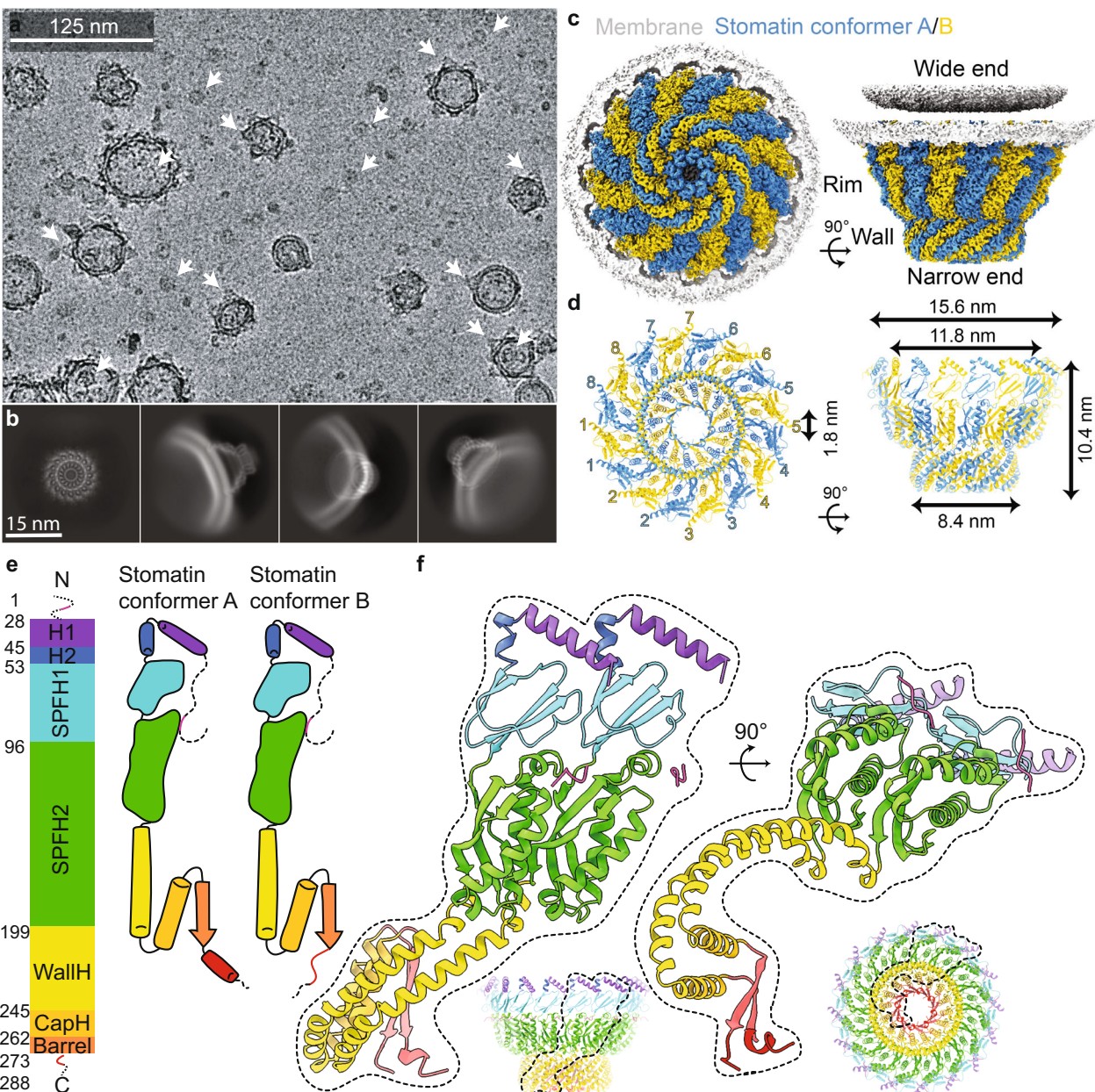

**Fig. 1 | Cryo-EM structure of the human Stomatin complex reveals a 16-subunit cage with alternating conformers. a** Cryo-EM micrograph of native membrane vesicles containing N-terminal ALFA-tagged human Stomatin. Arrows indicate particles showing top and side views of the cap-like complex. This experiment was independently repeated three times with similar results. **b** Selected 2D class averages reveal top and side orientations, showing that the Stomatin complex forms a closed, symmetric ring partially embedded in the membrane. **c** 3D reconstruction of the 16-mer Stomatin oligomer, displaying alternating subunit conformations (Conformer A: blue; Conformer B: yellow). Shown are bottom (left) and side (right) views, with the complex anchored to the membrane via its wide end and the narrow end forming a central opening. **d** Structural dimensions and subunit arrangement of the complex. Bottom view (left) shows the alternating A/B pattern with numbered subunits. Side view (right) indicates the overall height (~10.4 nm), external width (~15.6 nm), internal diameter (~11.8 nm), and pore diameter at the narrow end (~1.8 nm). **e** Domain architecture of Stomatin conformers. Both share the same topology: Flexible N-terminus (pink), two short hydrophobic helices (H1 (purple) and H2 (blue)), SPFH1 and SPFH2 domains (cyan and green), followed by a short wall helix (WallH, yellow), a cap helix (CapH, orange), and a C-terminal β-barrel (orange red). Conformer A ends in a short helix, while Conformer B adopts a looped C-terminal region (red). **f** Structures of Conformer A and B. Side (left) and bottom (right) views highlight the domain organization and asymmetry in the ring. Insets show the conformers within the full oligomeric cage (bottom center) and a top-down view from the narrow end, where the β-barrel forms a central pore.

(Fig. 3e). These salt bridges are repeated throughout the ring and form a conserved electrostatic network critical for oligomer stability. Sequence alignments show that many of these charged residues are conserved or replaced by similar amino acids in Podocin, STOML3, and MEC-2 (Supplementary Fig. 2). Importantly, several mutations at equivalent positions in Podocin are associated with nephrotic syndrome, further supporting the functional relevance of these contacts. These stabilizing electrostatic interactions define the overall ring architecture of the Stomatin complex. We next turned our attention to the inner structure at the narrow end to understand how Stomatin assembles into a stable homo-oligomer.

**Narrow end of the Stomatin complex forms a hydrophobic pore**
From the cytosolic view of the Stomatin complex (Fig. 4a), the oligomer forms a 16-stranded β-barrel, with each β-strand contributed by a separate subunit. The arrangement of these strands is unusual:

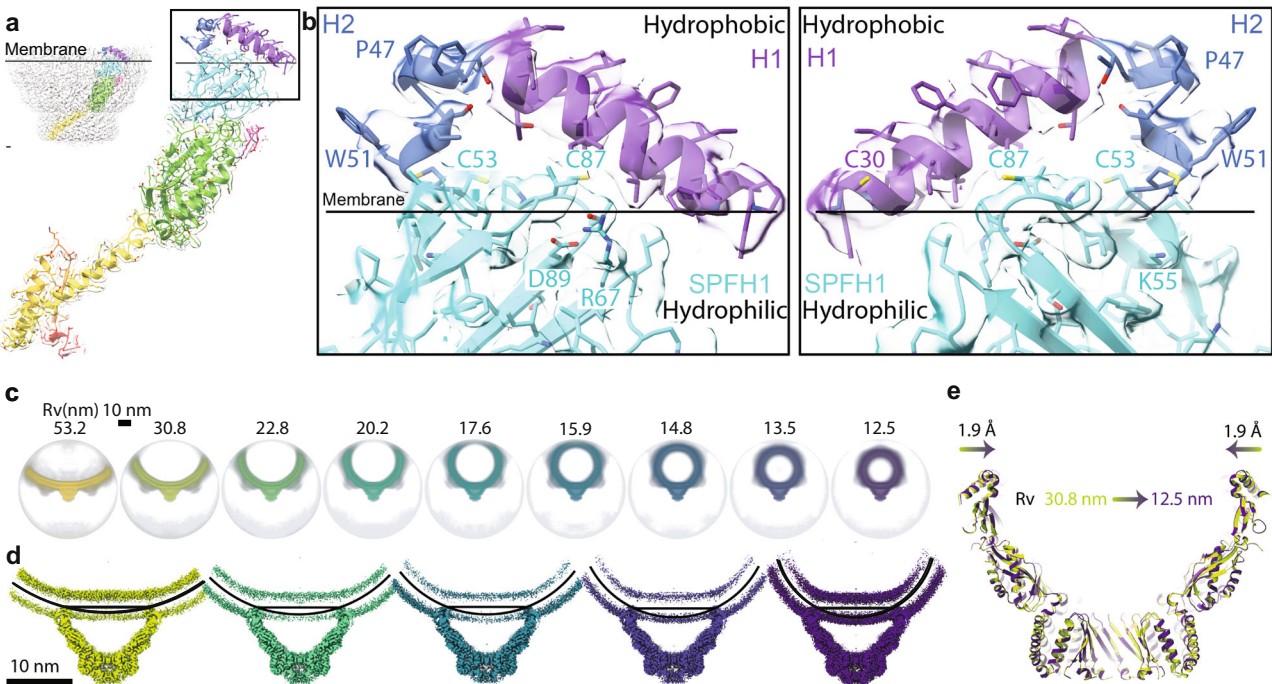

**Fig. 2 | Stomatin anchors to the membrane via N-terminal helices and the hydrophobic surface of SPFH1 and defines a mechanically distinct membrane domain. a** Side view of the cryo-EM density and atomic model of a Stomatin subunit showing its orientation relative to the membrane surface (solid line). Inset highlights the N-terminal helices (H1 and H2) and the membrane-facing hydrophobic region of the SPFH1 domain, both of which insert partially into the lipid bilayer. **b** Zoomed-in views of the membrane-anchoring interface. The N-terminal helices H1 and H2 (purple and blue) insert into the outer leaflet of the lipid bilayer, while the hydrophobic surface of the SPFH1 domain (cyan) also embeds into the membrane. The remaining surface of SPFH1 is hydrophilic and faces the cytosol, featuring stabilizing interactions such as a salt bridge between D89 and R67, and a positively charged residue, K55. Key residues involved in membrane interactions are labeled, including palmitoylation sites (C30, C53, C87), a membrane-anchoring

residue (W51), and the topology-influencing proline (P47). The right panel shows a 180° rotation around the Y-axis to reveal the inner side of the complex. **c** Cryo-EM 3D reconstructions of vesicles containing the Stomatin complex, sorted by vesicle radius of curvature (Rv), ranging from 53.2 nm to 12.5 nm. In all cases, the Stomatin complex maintains a consistent cap-like architecture. **d** Cross-sectional views of the reconstructions show that the membrane beneath the Stomatin complex remains flat, even as the rest of the vesicle exhibits curvature. Black curved lines represent the vesicle membrane, while straight lines mark the flattened microdomain under the complex. **e** Superimposed atomic models from vesicles of different curvature (Rv = 30.8 nm, green; Rv = 12.5 nm, purple) show only a ~ 1.9 nm change in the membrane-facing diameter on each side. This minimal variation indicates that the Stomatin complex enforces a rigid, curvature-resistant membrane microdomain.

although the sequence is identical, alternating subunits adopt different conformations. For instance, resides (e.g., F269) face outward in one subunit and inward in its neighbor (Fig. 4b). This organization gives rise to a narrow hydrophobic pore at the center of the barrel, with an internal diameter of approximately 1.8 nm. We also observed unassigned density near the hydrophobic core of the pore, which may dynamically regulate access. While the precise biological function of this pore remains to be determined, its position and chemical nature suggest it could act as a regulatory gate for molecular exchange within the membrane microdomain defined by the Stomatin complex. The structural transition between the cap helix and β-barrel domain begins at residues K263 and N264, which are similarly oriented in all subunits. However, starting at residue S265, side chain orientations begin to diverge between neighboring subunits, reflecting the alternating conformations and producing a β-barrel with C8 symmetry embedded within the C16 oligomer (Fig. 4c). The β-strand ends with two conserved prolines, P270 and P272, which act as kinks that direct the C-terminal regions in divergent paths: one subunit forms an α-helix that points into the pore, while the adjacent subunit adopts a loop that extends away. The final 8–10 C-terminal residues in both conformers are unresolved in the map, but their positions suggest that they could dynamically regulate pore diameter and selectivity—either blocking access or mediating passage of small molecules. Together, these results reveal that the narrow end of the Stomatin complex forms a highly ordered yet asymmetric hydrophobic pore, with alternating

subunit conformations shaping its unique architecture and potentially contributing to selective permeability.

## Discussion

### Structural insight into Stomatin assembly and membrane microdomain formation

Our cryo-EM structure of the human Stomatin complex in its native membrane environment reveals how Stomatin assembles into a 16-mer ring to define a mechanically distinct membrane microdomain. This structural insight builds on decades of biochemical studies exploring Stomatin's topology, cholesterol binding, palmitoylation, and oligomerization[5,20,21,23,25,26], but significantly revises previous models—particularly regarding the C-terminal region. Contrary to earlier assumptions, the C-terminus does not interact with the membrane; instead, it forms a β-barrel structure that plays a central role in oligomerization.

This cage-like architecture supports the long-standing concept of lipid rafts. Stomatin and other SPFH proteins, including Erlin[37], Flotillin[22], and Prohibitin[38], are widely considered markers of raft-like membrane domains. Their bacterial homologs, such as HlfK/C[39], FloA and FloT[40], have also been implicated in forming similar raft-associated structures. Cryo-electron tomography studies[41–43] have observed cage-like densities reminiscent of Stomatin and Flotillin structures, suggesting that this mechanism of membrane microdomain formation is evolutionarily conserved.

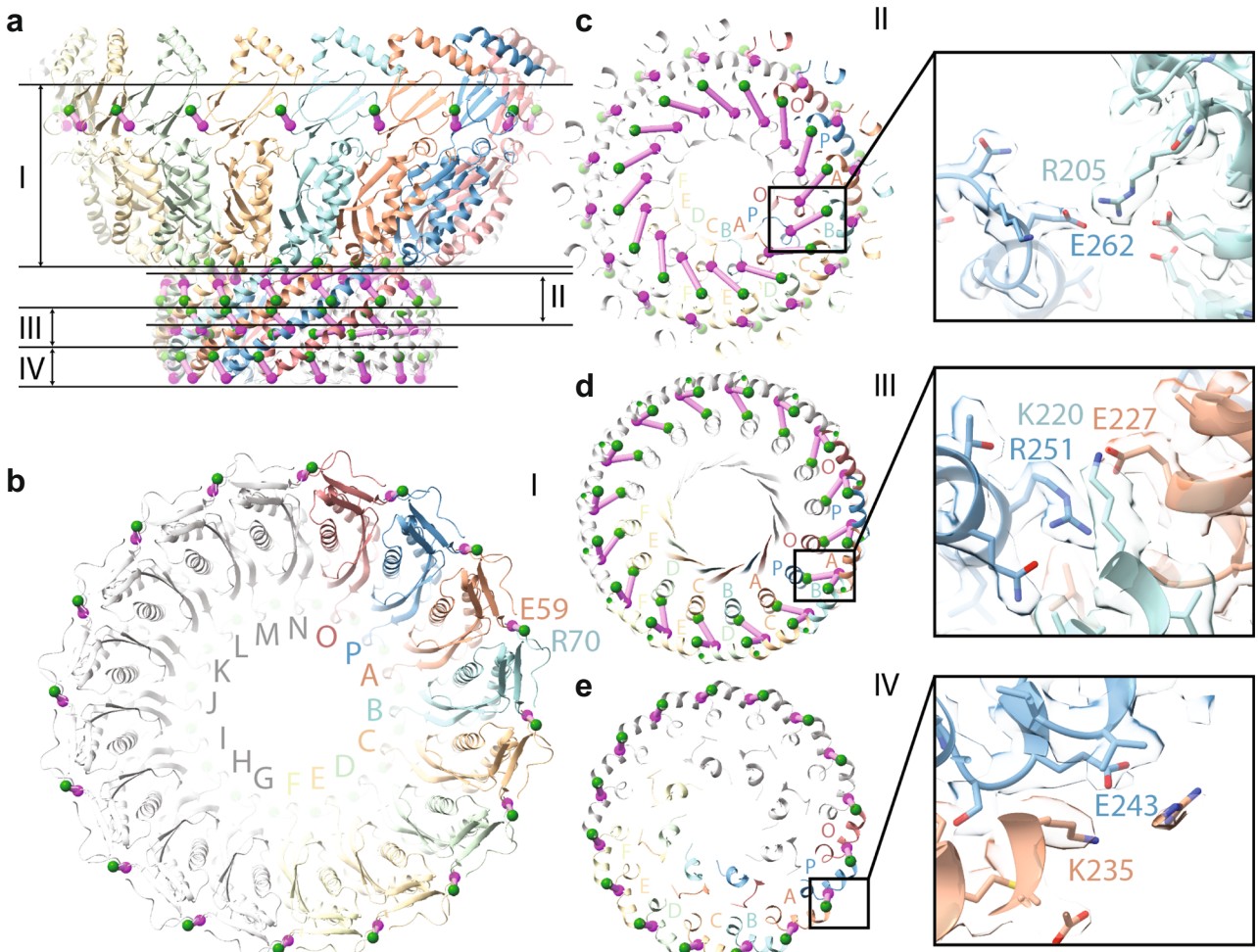

**Fig. 3 | Conserved salt bridges stabilize the Stomatin oligomer. a** Side view of the Stomatin 16-mer cage with subunits colored individually. The structure is divided into four layers (I–IV) based on height from the membrane interface. **b** Top-down view of the complex at layer I showing a salt bridge between E59 and R70 at the SPFH1 domain from neighboring subunits. **c** Layer II view showing a salt bridge between R205 (wall helix) and E262 (CapH) from subunits two positions apart, illustrating cross-subunit stabilization. **d** Layer III view highlighting interactions among E227 (wall helix), K220 (adjacent wall helix), and R251 (CapH from another subunit). **e** Layer IV view showing a salt bridge between E243 and K235 between neighboring subunits. Insets in (**b**–**e**) display zoomed-in views of representative salt bridges. Green and purple spheres mark residue positions. These salt bridges form a conserved electrostatic network critical for maintaining oligomer integrity and are enriched at positions associated with disease mutations in podocin.

## Membrane rigidity and mechanosensory function

Our data show that the Stomatin complex forms a flat, curvature-resistant membrane patch, even when embedded in vesicles of widely varying diameters. This rigidity arises from the oligomer's cage-like structure, which is anchored by short hydrophobic helices and the membrane-facing surface of the SPFH1 domain. Extensive inter-subunit interactions further stabilize the complex and allow it to locally resist membrane bending. These properties provide a mechanistic basis for the well-documented role of Stomatin family proteins in regulating mechanosensation and membrane tension. For example, MEC-2—a Stomatin homolog in *C. elegans*—enhances mechanosensory channel activity without direct binding to the channels themselves[9,14,15,44]. In the kidney, Podocin helps maintain slit diaphragm integrity and function under conditions of high filtration pressure[45,46]. STOML3 modulates Piezo sensitivity[17–19] and its deletion reduces light-touch responses in mice[8,17]. Small molecules like OB-1 disrupt STOML3 oligomerization and modulate Piezo activity[19]. Stomatin regulates ASIC channels, further linking it to sensory gating mechanisms[30,47]. While cholesterol binding has been proposed as a contributor to membrane stiffening, our structure does not reveal clear cholesterol densities. This suggests either transient interactions or enrichment of pre-ordered lipid domains, which warrants further investigation through targeted lipidomics.

## Functional compartmentalization and regulation

Beyond modulating membrane stiffness, Stomatin may serve as a scaffold for functional compartmentalization—enclosing regulatory targets within discrete membrane domains. Similar roles have been observed for other SPFH family proteins across biological systems. In bacteria, the HflK/C–FtsH complex forms a ring-like structure that encases the FtsH protease to regulate proteolysis[39,48]. In mitochondria, Prohibitin assembles into large oligomers that surround protease complexes, contributing to mitochondrial protein quality control. In *C. elegans*, the Stomatin homolog UNC-1 has been shown by cryo-ET to cap gap junctions, forming cage-like structures that likely modulate intercellular communication[41,42]. These precedents suggest that human Stomatin could similarly form caps over membrane-associated proteins—potentially regulating ion channels such as TRPC6 or ASICs. However, the internal diameter of the Stomatin complex may exclude larger proteins like Piezo1. In situ cryo-EM studies in mammalian cells will be essential to test whether Stomatin cages physically enclose signaling targets or organize them through lateral interactions within the membrane.

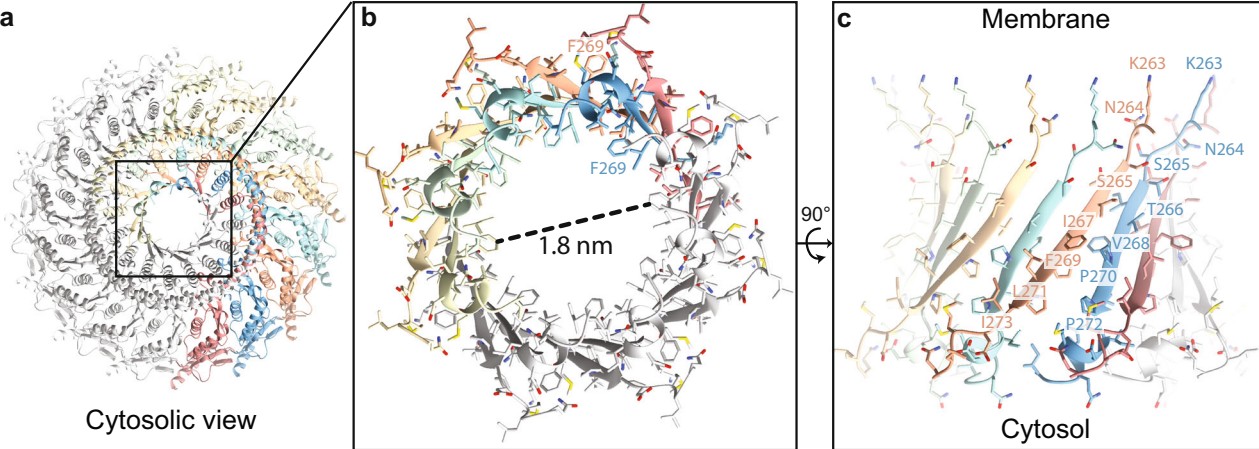

**Fig. 4 | Stomatin β-barrel domain forms a central hydrophobic portal at the narrow end. a** Cytosolic view of the Stomatin 16-mer showing the C-terminal β-barrel assembly at the narrow end. Each subunit contributes one β-strand, forming a closed circular structure. **b** Zoom-in of the β-barrel viewed from the cytosolic side. The central pore has an inner diameter of ~1.8 nm, suggesting possible selective diffusion of small molecules or ions. **c** Side view of the β-barrel structure rotated 90°, showing alternating strands from Conformer A and Conformer B. Conserved residues line the inner surface, forming a hydrophobic core. Residues from each conformer are color-coded and labeled to show side chain organization around the pore.

## Oligomeric flexibility across SPFH family proteins

A key question in the SPFH protein field is whether all members form static, closed oligomers like Stomatin and Flotillin, or whether some exist in both open and closed states depending on function. Structural studies of bacterial HflK/C have captured both conformations, even under native expression, suggesting functional switching. Similarly, Prohibitin has been observed in multiple oligomeric states, likely reflecting different biological roles or regulatory conditions. In contrast, our cryo-EM reconstructions of human Stomatin—determined from native vesicles—consistently show a stable, closed-ring architecture, supporting its role as a rigid membrane scaffold.

Interestingly, Stomatin forms a 16-mer homo-oligomer featuring a β-barrel structure at the C-terminus, similar to those seen in Flotillin and HflK/C. This architecture is achieved through two distinct conformations of the same sequence, alternating in the oligomer to enable β-barrel formation and generate internal C8 symmetry. We were also surprised by this result, as many SPFH family proteins—such as Flotillin1/2, Erlin1/2, and PHB1/2—are encoded in paralogous gene pairs, tend to co-localize, and form obligate hetero-oligomers. In contrast, Stomatin, Podocin, and STOML3 do not appear to have pairing partners. Instead, they seem to encode the capacity for functional complementation within a single sequence, using distinct conformations to achieve higher-order oligomerization. This may represent a generalizable mechanism for single-gene SPFH proteins to achieve architectural and functional complexity. Given the structural similarity in their core domains, it is plausible that Stomatin may form hetero-oligomers with closely related family members such as STOML3 and Podocin. These proteins share nearly identical lengths in the structured core, with most variation occurring at the N- and C-termini—regions unlikely to disrupt oligomer assembly. Understanding how oligomerization dynamics vary across SPFH proteins will require further structural analysis in physiologically relevant environments and may reveal diverse assembly mechanisms linked to their specialized cellular functions.

## Mechanistic link to disease

Many of the residues that help Stomatin proteins form oligomers and anchor to membranes are conserved and also found mutated in disease. For example, several disease-causing mutations in Podocin[33,35,36]—linked to nephrotic syndrome—occur at key structural sites in our Stomatin model (Supplementary Fig. 2). These mutations likely disrupt oligomerization, which may interfere with membrane organization and ion channel regulation. Our structure offers a mechanistic explanation for how such mutations cause disease and suggests potential therapeutic strategies. Stabilizing Podocin oligomers could help restore membrane integrity in kidney diseases, while disrupting oligomerization in other SPFH proteins—such as STOML3—might dampen mechanosensory responses and provide avenues for pain relief. For instance, the small molecule OB-1 has been shown to interfere with STOML3 assembly, though its precise mechanism remains unclear[19]. Given that Stomatin proteins also modulate ASIC channels, targeting their oligomeric state may offer strategies to treat neuroinflammation or influence brain activity. Ultimately, therapies that stabilize, disrupt, or reconfigure SPFH assemblies could reshape how we approach diseases linked to membrane microdomain dysfunction.

## Methods

### Cell culture

HEK293S GnTI⁻ cells (ATCC CRL-3022) were cultured in Freestyle 293 medium supplemented with 2% fetal bovine serum at 37 °C.

### Construct design

A DNA sequence encoding a GGGS linker followed by the ALFA tag peptide (PSRLEEELRRRL) was inserted immediately after the start codon of human Stomatin (UniProt: P27105-1) into the pEG BacMam vector for expression.

### Expression and Stomatin vesicle preparation

Baculoviruses carrying human Stomatin were generated and amplified through two rounds in *Spodoptera frugiperda* (Sf9) cells. Suspension cultures of HEK293S GnTI⁻ cells were grown at 37 °C and infected with 10% (v/v) virus at a density of ~3 × 10⁶ cells/mL. Sixteen hours post-infection, 10 mM sodium butyrate was added to induce expression, and cells were shifted to 30 °C. Cells were harvested 48 h after induction.

Frozen cell pellets from 4 L of culture were thawed and resuspended in 160 mL lysis buffer (20 mM K-HEPES pH 7.4, 300 mM KCl, 0.5 mM MgCl₂, 5 mM DTT), supplemented with protease inhibitors (10 μg/mL leupeptin, 10 μg/mL pepstatin A, 1 mM benzamidine, 2 μg/mL aprotinin, 0.3 mg/mL AEBSF) and nucleases (~100 μg/mL DNase I and RNase A). Cells were homogenized using a Dounce homogenizer (30–60 strokes), transferred to a metal beaker, and sonicated on ice using a probe sonicator (60% power, four 30 s pulses with 30 s cooling intervals).

The homogenate was centrifuged at 12,000 × g for 10 min twice. Membrane vesicles from the supernatant were incubated for 1 h at room temperature with 1 mL ALFA Selector CE resin (NanoTag) pre-equilibrated in buffer B (20 mM K-HEPES pH 7.4, 300 mM KCl, 5 mM DTT, 10% glycerol). The resin was batch-washed twice with ~20 mL buffer B, collected by centrifugation at 1000 × g for 1 min, and washed again with 15 mL buffer B. The resin was transferred to a gravity column and washed sequentially with 5 mL buffer B, followed by 15 mL buffer B lacking glycerol. Bound vesicles were eluted in three steps using buffer B containing 0.2 mM ALFA peptide: two washes of 5 column volumes (CV) and one wash of 3 CV, each incubated for 30 min at room temperature. Eluted fractions were pooled, kept on ice, and concentrated to an $OD_{280}$ of ~1.2 using a 2 mL Amicon concentrator (100 kDa cutoff).

## Grid preparation and data collection
Quantifoil R1.2/1.3 400 mesh holey carbon gold grids were glow-discharged for 30 s. Three microliters of concentrated vesicles were applied to freshly glow-discharged grids and incubated for 3 min at 22 °C under 100% humidity. Grids were blotted manually from the edge using filter paper. A second 3 μL sample was applied, followed by blotting in a Leica GP2 for 8 s after 30 s incubation. Grids were vitrified in liquid ethane (−180 °C) and stored in liquid nitrogen until imaging.

Data were collected on a 300 keV Titan Krios equipped with a Gatan K3 Summit camera and energy filter (20 eV slit). A calibrated pixel size of 0.844 Å was used.

A total of 9026 movies were acquired in super-resolution mode (pixel size: 0.422 Å), with a defocus range of −0.3 to −2.3 μm. Each movie was recorded for 2 s with 50 frames, a dose rate of 28.08 e$^-$/Å$^2$/s, and a total dose of 56.16 e$^-$/Å$^2$.

## Cryo-EM data processing for Stomatin vesicles
Movies were motion-corrected and CTF parameters estimated using patch-based workflows in cryoSPARC v4.2.0[49]. An initial 500 particles were manually picked to train a Topaz model, which was then used for automated picking. A single round of 2D classification identified high-quality particles. An ab initio model was generated with C16 symmetry, and 1,157,242 particles were selected for refinement.

Homogeneous refinement (C16 symmetry) yielded a 2.9 Å map. Discontinuity at the narrow end suggested symmetry breaking, prompting 3D classification with a focused mask and C8 symmetry. Two major classes related by ~22.5° rotation were identified. Volume alignment aligned one class to the other, enabling particle merging. Local refinement (C8 symmetry) improved resolution to 2.6 Å. Reference-based motion correction and global CTF refinement further improved resolution to 2.25 Å. Final local CTF refinement with Ewald sphere correction on 698,428 particles yielded a 2.2 Å map; a final refinement reached 2.17 Å by gold-standard FSC = 0.143. Local resolution estimation was performed in cryoSPARC.

## Model building, refinement, and analysis
The AlphaFold-predicted structure of human Stomatin was fitted into the 2.2 Å map using molecular dynamics flexible fitting in ISOLDE[50]. Two manually placed subunits were positioned in Coot[51] and expanded via noncrystallographic symmetry to complete the model. Iterative refinement was conducted in Coot and Phenix[52] real-space refinement. Regions of weak density were omitted. The final model includes residues 14–18 and 29–278. Model visualization and figure generation were performed in UCSF ChimeraX[53].

## Reporting summary
Further information on research design is available in the Nature Portfolio Reporting Summary linked to this article.

## Data availability
The cryo-EM density map generated in this study has been deposited in the Electron Microscopy Data Bank (EMDB) under the following accession code: EMD-70485. The corresponding atomic coordinates has been deposited in the Protein Data Bank (PDB) under accession code 9OH9.

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

## Acknowledgments

This work was supported by startup funding from the McDonnell Center for Cellular and Molecular Neurobiology, the Department of Cell Biology & Physiology, and the Center for the Investigation of Membrane Excitability Diseases at Washington University in St. Louis. Experiments, data acquisition, and analyses were performed in part through the use of the Washington University Center for Cellular Imaging (WUCCI), supported by Washington University School of Medicine, the Children's Discovery Institute of Washington University and St. Louis Children's Hospital (CDI-CORE-2015-505 and CDI-CORE-2019-813), and the Foundation for Barnes-Jewish Hospital (3770 and 4642). Some of this work was performed at the National Center for CryoEM Access and Training (NCCAT) and the Simons Electron Microscopy Center located at the New York Structural Biology Center, supported by the NIH Common Fund Transformative High Resolution Cryo-Electron Microscopy program (U24 GM129539 and NIGMS R24 GM154192), and by grants from the Simons Foundation (SF349247) and the New York State Assembly. This work was also supported by a Pilot and Feasibility grant (CIMED-25-01) from the Center for the Investigation of Membrane Excitability Diseases.

## Author contributions

J.S. and Z.F. cloned, expressed, and purified the Stomatin complex for cryo-EM, prepared cryo-EM samples, collected data, processed images, built atomic models, and performed all structural analyses. Z.F. provided mentorship and guidance on sample preparation and image processing. J.S., S.L., and Z.F. wrote the manuscript and prepared the

figures. Z.F. and S.L. conceived and supervised the project. All authors contributed to manuscript editing.

## Competing interests

All authors declare no competing interests.
