## [Transparent Peer Review file · Nature Communications]

Structural basis for membrane microdomain formation by a human Stomatin complex

Corresponding Author: Dr Ziao Fu

Version 0:

Reviewer comments:

Reviewer #1

(Remarks to the Author)
NCOMMS-25-31050
Review comments

Authors has first reported cryo-EM structure of human stomatin at 2.2 angstrom resolution. Authors describes precise structural features on stomatin, in particular forming a 16-subunit ring-shaped homo-oligomer, accommodating membrane architecture, via many conserved inter-subunit salt bridges stabilizing the oligomer. Authors also indicate that the structure also explains corresponding Podocin mutation involved in nephrotic syndrome. Data of cryo-EM and the description of the structure in the text is clear. Many readers and I will be interested in this article.

Major points:

1. Authors discuss the sizes of oligomeric structure are ranging from 53.2 nm to 12.5 nm in Figure 2c (line 547). I cannot understand the mechanism. How is the oligomer change the molecular size ? Does the structure keep the oligomeric state 16-mer, or form different oligomers ? Please explain more precisely. I think that this is very interesting point.
2. In the cytosolic side, the central pore has an inner diameter of ~1.8 nm, very small. I hope authors may explain the meaning of this small pore.

Minor points:

Lines 151, 154, and 161: Figure numbers are wrong. Figs. 3a, 3b, and 3c may be revised to Figs. 4a, 4b, and 4c.

Line 525 in Fig. 1f: I think "Models" of Conformer A and B are not correct, instead "Structures" are correct.

Reviewer #2

(Remarks to the Author)

Stoner et al reported a high-resolution structure of human Stomatin complex. The authors over-expressed Stomatin in mammalian cells and isolated Stomatin complex on vesicles to high purity, followed by high-resolution cryo-EM SPA. This is the first high-resolution structure of Stomatin, therefore reconcile the long-term biochemical property, such as cholesterol binding, oligomerization etc. The structural analysis is performed professionally with appropriate interpretations. The methodology is well described to detail. I have the following comments for the authors to consider before acceptance to publish.

1. It is quite unexpected that Stomatin forms a C8 symmetry instead of C16; is such symmetry break a general theme in this family proteins? In the whole dataset, any other symmetry was identified?
2. What are the potential candidates that has been studied to be associated with the Stomatin complex? In this purified Stomatin vesicles, any potential interacting partners is identified by mass spectrometry?
3. How many Stomatin on each vesicle? From Figure 1a and Figure 2c there appears to be extra Stomatin cones in the averaged map? Are there molecular interactions among Stomatins oligomers?

4. How the disease-related mutations on the homologs contribute to Stomatin cage-like structure assembly? The interface mutations should be biochemically analyzed to assess their contribution to oligomer formation.

5. The authors primarily focused on protein structure determination and analysis of structural features in this work. They described Stomatin's hydrophobic interaction with the membrane via an N-terminal helix and identified key interacting residues (e.g., C30, C53, C87, R67, D89). Similarly, they meticulously analyzed interaction details within the N-terminal 16-mer (salt bridges involving R205, K220, E227, R251, and E262) and the C-terminal 8-mer (salt bridge between K235 and E243). Did the authors perform any mutations at these sites to further validate their functional significance in assembly?

6. It appears that Stomatin binds to cholesterol-rich membrane? It would be informative to measure the lipid species in these Stomatin-enrich vesicles compared with other membrane, since this sample is purified to high purity?

7. Most of the sequences can be built into the map (L326). Any other roles in these disordered/unresolved regions? How conserved are these regions? Do they contribute to membrane binding/recognition?

8. The authors directly utilized purified vesicles to localize and resolve Stomatin structure, preserving its native state. Did the authors attempt to purify monomeric Stomatin protein and subsequently reconstitute it onto liposomes to assess whether it can adopt a conformation similar to the native state and validate its function? This may help to understand the dynamic assembly process.

Reviewer #3

(Remarks to the Author)

Reviewer #4

(Remarks to the Author)

I think, this work would make a prominent contribution to this and related scientific fields. In other words, these findings provide a molecular framework for understanding how Stomatin and its homolog organize membrane architecture, and for understanding the molecular mechanism of another family protein, Podocin which intimately relates to nephrotic syndrome. I think, most excellent result of this work is the determination of the high-resolution cryo-EM structure of full-length human Stomatin in its membrane environment. Stomatin assembles into a closed 16-subunit ring. The structure shows how alternating protein conformers form a rigid scaffold through conserved inter-subunit salt bridges and a C-terminal beta-barrel, generating a hydrophobic pore at the narrow end of the complex.

The methodology used in this study was sound and there were no flaws in the data analysis, interpretation, or conclusions.

Research methods contain sufficient detail to ensure reproducibility.

I think, this study well satisfy the standards expected in these scientific fields.

Comments to Author,

1. line 144, the sentence "residues are conserved in Podocin,STOML3,---" should be changed more properly to the sentence "residues are conserved or replaced by similar amino acids in Podocin,STOML3,---".

2. line 184, the sentence "that this mechanism of membrane microdomain formation may be evolutionarily conserved" might be changed to the sentence "that this mechanism of membrane microdomain formation is evolutionarily conserved".

Version 1:

Reviewer comments:

Reviewer #1

(Remarks to the Author)

I accepted all the revised points.

Reviewer #2

(Remarks to the Author)

The authors have provided point-to-point notes to each of the questions I have asked and adequately addressed the

questions. This is a very interesting and concrete study of Stomatin complex by cryo-EM. I suggest to accept the manuscript for publication and have no further comments. Thank you.

Reviewer #3

(Remarks to the Author)

Reviewer #4

(Remarks to the Author)

The reviewers' questions and comments have been appropriately answered by authors, and the manuscript has been revised adequately.

REVIEWER COMMENTS

Reviewer #1 (Remarks to the Author):

NCOMMS-25-31050

Review comments

Authors has first reported cryo-EM structure of human stomatin at 2.2 angstrom resolution. Authors describes precise structural features on stomatin, in particular forming a 16-subunit ring-shaped homo-oligomer, accommodating membrane architecture, via many conserved inter-subunit salt bridges stabilizing the oligomer. Authors also indicate that the structure also explains corresponding Podocin mutation involved in nephrotic syndrome. Data of cryo-EM and the description of the structure in the text is clear. Many readers and I will be interested in this article.

We thank the reviewer #1 for the positive and encouraging assessment of our work. We are glad that the structural insights and biological implications of the Stomatin complex were found to be clear and of interest. We appreciate your thoughtful feedback and support for the manuscript.

Major points:

1. Authors discuss the sizes of oligomeric structure are ranging from 53.2 nm to 12.5 nm in Figure 2c (line 547). I cannot understand the mechanism. How is the oligomer change the molecular size ? Does the structure keep the oligomeric state 16-mer, or form different oligomers ? Please explain more precisely. I think that this is very interesting point.

We thank the reviewer for highlighting this important point. We apologize for the confusion and have revised the text (Lines 113–121) to clarify the interpretation.

To clarify: the oligomeric state of Stomatin remains a stable 16-mer ring across all vesicles analyzed. The reported range of sizes (53.2 nm to 12.5 nm) refers not to changes in the Stomatin complex itself, but to the **diameter of the vesicles** containing the complex. In other words, the Stomatin oligomer does **not** change size or stoichiometry. Instead, it consistently forms a rigid, cap-like 16-mer that **locally flattens** the underlying membrane, even when the surrounding vesicle curvature varies widely.

We have now explicitly stated that the structural integrity and oligomeric state of the Stomatin complex is maintained across all vesicle sizes, and that the small variations in the membrane-facing footprint (~1.9 Å) reflect slight adjustments at the protein-lipid interface rather than a change in oligomeric assembly. We believe this rigidity is a key feature of how Stomatin defines a mechanically distinct microdomain within the membrane.

Original text (Lines 113–115)

“Using cryo-EM, we found that vesicles containing Stomatin exhibited a wide range of radii—from ~53 nm to ~12.5 nm (Fig. 2c). Despite this variation, the membrane curvature directly beneath the Stomatin complex remained constant, while surrounding membrane conformed to the vesicle’s overall curvature.”

Revised text

“Using cryo-EM, we found that vesicles containing Stomatin exhibited a wide range of radii—from ~53 nm to ~12.5 nm (Fig. 2c). **Importantly, the Stomatin complex itself maintained a consistent 16-mer oligomeric architecture across all vesicles.** Despite this variation, the membrane curvature directly beneath the Stomatin complex remained constant, while surrounding membrane conformed to the vesicle’s overall curvature.”

2. In the cytosolic side, the central pore has an inner diameter of ~1.8 nm, very small. I hope authors may explain the meaning of this small pore.

Response:

We thank the reviewer for this important question. We agree that the small central pore (~1.8 nm) is a very interesting feature of the Stomatin complex and have added further discussion to clarify its potential significance.

Original text (Lines 154–157):

“This organization gives rise to a narrow hydrophobic pore at the center of the barrel, with an internal diameter of approximately 1.8 nm. Additional unassigned density is present near the hydrophobic region of the pore which could potentially completely block the pore.”

Revised text (Lines 154–165):

“This organization gives rise to a narrow hydrophobic pore at the center of the barrel, with an internal diameter of approximately 1.8 nm. **We also observed unassigned density near the hydrophobic core of the pore, which may dynamically regulate access. While the precise biological function of this pore remains to be determined, its position and chemical nature suggest it could act as a regulatory gate for molecular exchange within the membrane microdomain defined by the Stomatin complex.**”

Minor points:

Lines 151, 154, and 161: Figure numbers are wrong. Figs. 3a, 3b, and 3c may be revised to Figs. 4a, 4b, and 4c.

Response:

We thank the reviewer for catching this oversight. The figure citations in Lines 151, 154, and 161 have been corrected to Figs. 4a, 4b, and 4c in the revised manuscript.

Line 525 in Fig. 1f: I think “Models” of Conformer A and B are not correct, instead “Structures” are correct.

Response:

We appreciate the reviewer’s suggestion. We agree that “Structures” is a more accurate term than “Models” in this context and have updated the label in Fig. 1f and the corresponding legend accordingly.

Original text (Line 525, Fig. 1f legend):

“Models of Conformer A and B. Side (left) and bottom (right) views highlight the domain organization and asymmetry in the ring.”

Revised text:

“**Structures** of Conformer A and B. Side (left) and bottom (right) views highlight the domain organization and asymmetry in the ring.”

Reviewer #2 (Remarks to the Author):

Stoner et al reported a high-resolution structure of human Stomatin complex. The authors over-expressed Stomatin in mammalian cells and isolated Stomatin complex on vesicles to high purity, followed by high-resolution cryo-EM SPA. This is the first high-resolution structure of Stomatin, therefore reconcile the long-term biochemical property, such as cholesterol binding, oligomerization etc. The structural analysis is performed professionally with appropriate interpretations. The methodology is well described to detail. I have the following comments for the authors to consider before acceptance to publish.

Response:

We thank the reviewer #2 for the thoughtful and encouraging assessment of our work. We are pleased that the structural analysis, interpretations, and methodology were found to be professional and well-executed. We appreciate the constructive comments and address each of them in detail below.

1. It is quite unexpected that Stomatin forms a C8 symmetry instead of C16; is such symmetry break a general theme in this family proteins? In the whole dataset, any other symmetry was identified?

Response:

We thank the reviewer for this insightful comment. We were also surprised by the local symmetry break observed in the Stomatin complex. While the overall oligomer forms a C16-symmetric ring, the C-terminal β -barrel region adopts local C8 symmetry due to alternating conformations of identical subunits (Conformer A and B).

This alternating conformation strategy appears to be a distinct mechanism used by certain SPFH family members. Notably, other complexes such as Flotillin1/2, Erlin1/2, and PHB1/2 are typically encoded as paralogous gene pairs, are co-expressed, and often form obligate hetero-oligomers. In contrast, proteins like Stomatin, Podocin, and STOML3 do not have paralogs that form stable complexes with them. Instead, they appear to achieve higher-order assembly by encoding two conformers within the same sequence, which complement one another structurally within the oligomer.

In our dataset, we did not observe any other oligomeric symmetries. All particles converged on a stable 16-mer architecture with consistent structural features across 2D classes and 3D reconstructions. The symmetry breaking is thus confined to the C-terminal pore region, and not a result of mixed stoichiometries.

We believe this modular asymmetry may represent a broader principle among SPFH proteins—either achieved by hetero-oligomerization (e.g., Flotillin, Erlin) or by alternating conformations of a single sequence (e.g., Stomatin). We have revised the Results (Lines 68–70, 158–159) and Discussion (Lines 227–229) to clarify these points and highlight the possible evolutionary divergence in oligomerization strategies among SPFH family members.

Original: (Lines 68–70)

The ring is made of two alternating forms of the protein, called Conformer A and Conformer B, arranged in a repeating A–B–A–B pattern.

Revised:

The ring is made of two alternating forms of the protein, called Conformer A and Conformer B, arranged in a repeating A–B–A–B pattern, **resulting in local C8 symmetry within the β -barrel region despite the overall C16 ring architecture.**

Original: (Lines 158–159)

...side chain orientations begin to diverge between neighboring subunits, reflecting the alternating conformations (Fig. 3c).

Revised:

...side chain orientations begin to diverge between neighboring subunits, reflecting the alternating conformations **and producing a β -barrel with C8 symmetry embedded within the C16 oligomer** (Fig. 4c).

Original: (Lines 227–229)

This architecture is achieved through two distinct conformations of the same sequence, alternating in the oligomer to enable β -barrel formation.

Revised:

This architecture is achieved through two distinct conformations of the same sequence, alternating in the oligomer to enable β -barrel formation and generate internal C8 symmetry. We were also surprised by this result, as many SPFH family proteins—such as Flotillin1/2, Erlin1/2, and PHB1/2—are encoded in paralogous gene pairs, tend to co-localize, and form obligate hetero-oligomers. In contrast, Stomatin, Podocin, and STOML3 do not appear to have pairing partners. Instead, they seem to encode the capacity for functional complementation within a single sequence, using distinct conformations to achieve higher-order oligomerization. This may represent a generalizable mechanism for single-gene SPFH proteins to achieve architectural and functional complexity.

2. What are the potential candidates that has been studied to be associated with the Stomatin complex? In this purified Stomatin vesicles, any potential interacting partners is identified by mass spectrometry?

Response:

We thank the reviewer for this thoughtful question. Identifying Stomatin-interacting partners is a high priority for future work, particularly in the context of membrane signaling and ion channel regulation. In the current study, our primary focus was to determine the high-resolution structure of the human Stomatin complex using a purified vesicle system. Although prior studies have reported potential Stomatin-associated proteins, such as junctional components, those analyses were beyond the scope of this work. We agree that identifying physiological interaction partners remains an important direction for future investigation.

3. How many Stomatin on each vesicle? From Figure 1a and Figure 2c there appears to be extra Stomatin cones in the averaged map? Are there molecular interactions among Stomatins oligomers?

Response:

We thank the reviewer for this thoughtful question. The number of Stomatin complexes per vesicle varies depending on vesicle size and preparation, and some vesicles do appear to contain more than one copy. Based on rough estimations from our current 2D classification results, vesicles can carry one to several (more than 16) Stomatin complexes. However, since these vesicles were derived from an overexpression system, the protein density may not reflect physiological levels. A more accurate assessment will require cryo-electron tomography and analysis of vesicles derived from native tissues, which we are actively pursuing.

Regarding potential interactions between Stomatin oligomers, this is a key area of interest for us. The high copy number per complex and the presence of an unstructured N-terminus suggest the possibility of multivalent interactions, potentially mediated by weak intermolecular contacts. However, in our current structural dataset, we did not observe direct or ordered interactions between adjacent Stomatin complexes, and no such interfaces were resolved. We

look forward to exploring these potential interactions further using complementary structural and functional approaches in future work.

4. How the disease-related mutations on the homologs contribute to Stomatin cage-like structure assembly? The interface mutations should be biochemically analyzed to assess their contribution to oligomer formation.

Response:

We appreciate the reviewer's important point regarding disease-related mutations and their potential effects on Stomatin cage assembly. In this study, we focused on structural determination of wild-type human Stomatin and did not perform mutational or biochemical analysis of specific interface residues. However, our structure reveals that several residues involved in oligomerization and membrane anchoring—particularly those forming inter-subunit salt bridges or interacting with the lipid bilayer—correspond to disease-linked mutations in Podocin, a Stomatin homolog.

Interestingly, Stomatin itself is not known to carry disease-associated mutations, likely due to its ubiquitous expression across all cell types and developmental stages, making loss-of-function mutations potentially embryonically lethal or nonviable. In contrast, Podocin is exclusively expressed in the kidney, and mutations at equivalent residues have been strongly linked to steroid-resistant nephrotic syndrome. These distinctions highlight the biological relevance of our structural mapping and underscore the value of investigating how such mutations impact oligomer formation and membrane organization.

We fully agree that biochemical validation of interface mutants, particularly those mimicking disease mutations in **Podocin**, represents a critical next step. This is an important future direction of our lab, and we are currently developing reagents to analyze how disease-linked mutations affect complex formation, stability, and function.

5. The authors primarily focused on protein structure determination and analysis of structural features in this work. They described Stomatin's hydrophobic interaction with the membrane via an N-terminal helix and identified key interacting residues (e.g., C30, C53, C87, R67, D89). Similarly, they meticulously analyzed interaction details within the N-terminal 16-mer (salt bridges involving R205, K220, E227, R251, and E262) and the C-terminal 8-mer (salt bridge between K235 and E243). Did the authors perform any mutations at these sites to further validate their functional significance in assembly?

Response:

We thank the reviewer for recognizing the detailed structural analysis and for raising this important question. In this study, our focus was to define the high-resolution structure of the wild-type human Stomatin complex in a native membrane environment. As such, we did not perform site-directed mutagenesis or functional assays to validate the role of specific residues in oligomerization or membrane interaction.

However, the residues highlighted in our structural analysis—including C30, C53, C87, R67, D89, and those forming the inter-subunit salt bridge network—are all highly conserved and map precisely to the oligomerization and membrane-binding interfaces. Many of these positions

correspond to disease-associated mutations in Podocin, strongly suggesting their functional importance.

We fully agree that validating these sites biochemically and functionally is a critical next step. We are currently generating a panel of targeted mutants to examine their roles in **Podocin** complex assembly, membrane interaction, and functional stability.

6. It appears that Stomatin binds to cholesterol-rich membrane? It would be informative to measure the lipid species in these Stomatin-enrich vesicles compared with other membrane, since this sample is purified to high purity?

Response:

We appreciate the reviewer's insightful suggestion. We are actively moving toward this direction—investigating the lipid composition of Stomatin-enriched membranes, including potential cholesterol association. However, we would like to note that the current system is based on overexpression, and the lipid environment surrounding the complex may not reflect the native lipid composition associated with endogenous Stomatin. Therefore, any lipid enrichment observed in this context must be interpreted with caution. Determining cholesterol association and overall lipid selectivity is an important goal of our future work. We are particularly interested in exploring whether SPFH family members—including Stomatin, Podocin, and STOML3—share conserved mechanisms for lipid preference or selection, which may contribute to their ability to define membrane microdomains. We believe that a more physiologically relevant approach will involve CRISPR-based knock-in tagging of endogenous loci to preserve native expression levels, combined with lipidomics and comparative analysis of wild-type, knock-in, and knockout backgrounds. This will enable a more accurate and comprehensive understanding of protein-lipid interactions across the family, beyond the limits of overexpression systems.

7. Most of the sequences can be built into the map (L326). Any other roles in these disordered/unresolved regions? How conserved are these regions? Do they contribute to membrane binding/recognition?

Response:

We thank the reviewer for this thoughtful question. In our structure, the majority of the Stomatin sequence was well resolved, with the exception of two flexible regions, in the N-terminal cytosolic domain and the extreme C-terminal tail. These unresolved regions are generally predicted to be intrinsically disordered and exhibit relatively low sequence conservation across the SPFH family. The N-terminal region preceding the membrane-interacting helices could participate in modulating oligomer clustering through weak multivalent interactions, or in recruiting accessory proteins. Investigating the functional role of these regions remains an active area of interest in our lab.

8. The authors directly utilized purified vesicles to localize and resolve Stomatin structure, preserving its native state. Did the authors attempt to purify monomeric Stomatin protein and subsequently reconstitute it onto liposomes to assess whether it can adopt a conformation similar to the native state and validate its function? This may help to

understand the dynamic assembly process.

Response:

This is a great suggestion. We are also very curious about how Stomatin and other SPFH assemblies form on membrane surfaces, and we are actively pursuing this direction. In the present study, we prioritized preserving the native oligomeric state of Stomatin, which is often disrupted by detergent-based purification. Therefore, we chose to work with vesicle-associated complexes to retain their native assembly and membrane interactions. While we have not yet attempted to purify monomeric Stomatin followed by reconstitution onto liposomes, we recognize this approach could provide valuable insight into the dynamic assembly process.

Reviewer #3 (Remarks to the Author):

Reviewer #4 (Remarks to the Author):

I think, this work would make a prominent contribution to this and related scientific fields. In other words, these findings provide a molecular framework for understanding how Stomatin and its homologs organize membrane architecture, and for understanding the molecular mechanism of another family protein, Podocin which intimately relates to nephrotic syndrome. I think, most excellent result of this work is the determination of the high-resolution cryo-EM structure of full-length human Stomatin in its membrane environment. Stomatin assembles into a closed 16-subunit ring. The structure shows how alternating protein conformers form a rigid scaffold through conserved inter-subunit salt bridges and a C-terminal beta-barrel, generating a hydrophobic pore at the narrow end of the complex.

The methodology used in this study was sound and there were no flaws in the data analysis, interpretation, or conclusions.

Research methods contain sufficient detail to ensure reproducibility.

I think, this study will satisfy the standards expected in these scientific fields.

Response:

We sincerely thank the reviewer for their thoughtful and encouraging comments. We are very pleased that the reviewer finds our structural analysis of the Stomatin complex to be rigorous and impactful. We agree that this work lays a structural foundation for understanding how Stomatin and its homologs contribute to membrane organization, and we are particularly excited about the implications this has for elucidating the function of Podocin in nephrotic syndrome. We appreciate the reviewer's recognition of the methodological rigor and clarity in data presentation,

and we will continue to explore these mechanisms in future work.

Comments to Author,

1. line 144, the sentence "residues are conserved in Podocin,STOML3,---" should be changed more properly to the sentence "residues are conserved or replaced by similar amino acids in Podocin,STOML3,---".

Response:

We thank the reviewer for the helpful suggestion. We have revised the sentence at line 144 to: **“These residues are conserved or replaced by similar amino acids in Podocin, STOML3, and other homologs.”**

Original sentence (Line 144):

“These residues are conserved in Podocin, STOML3, and MEC-2.”

Revised sentence:

“These residues are conserved **or replaced by similar amino acids** in Podocin, STOML3, and MEC-2.”

2. line 184, the sentence "that this mechanism of membrane microdomain formation may be evolutionarily conserved" might be changed to the sentence" that this mechanism of membrane microdomain formation is evolutionarily conserved".

Response:

We thank the reviewer for this suggestion. We agree that the revised wording strengthens the statement and better reflects the structural and evolutionary similarities observed. Accordingly, we have revised the sentence at line 184 to:

Original sentence (Line 184):

“...suggesting that this mechanism of membrane microdomain formation may be evolutionarily conserved.”

Revised sentence:

“...suggesting that this mechanism of membrane microdomain formation **is** evolutionarily conserved.”